# Meltwater sediment transport as the dominating process in mid-latitude trough mouth fan formation

Benjamin Bellwald [1✉], Sverre Planke[1,2,3], Lukas W. M. Becker [4] & Reidun Myklebust[5]

Trough mouth fans comprise the largest sediment deposits along glaciated margins, and record Pleistocene climate changes on a multi-decadal time scale. Here we present a model for the formation of the North Sea Fan derived from detailed horizon and attribute interpretations of high-resolution processed 3D seismic reflection data. The interpretation shows that stacked channel-levee systems form up to 400 m thick sedimentary sequences. The channels are elongated and can be traced from the shelf edge towards the deep basin for distances of >150 km, and document long-distance sediment transport in completely disintegrated water-rich turbidite flows. Downslope sediment transport was a continuous process during shelf-edge glaciations, reaching accumulation rates of 100 m/kyr. Our data highlight that exceptionally large volumes of meltwater may discharge to the slopes of trough mouth fans and trigger erosive turbidite flows. We conclude that freshwater supply is likely an underestimated factor for sedimentary processes during glacial cycles.

[1] Volcanic Basin Petroleum Research AS (VBPR), Oslo, Norway. [2] Centre for Earth Evolution and Dynamics (CEED), University of Oslo, Oslo, Norway. [3] Research Centre for Arctic Petroleum Exploration (ARCEx), UiT The Arctic University of Norway, Tromsø, Norway. [4] Department of Earth Science, University of Bergen, Bergen, Norway. [5] TGS, Asker, Norway. ✉email: benjamin@vbpr.no

Trough mouth fans are products of repeated glacigenic sediment delivery from former fast-flowing outlets of ice sheets, and act as high-resolution paleoclimate and ice-sheet monitors[1–3]. The fans have highest sedimentation rates and maximum periods of growth during glacial maxima, whereas they become ice-distal glacimarine environments with low sedimentation rates during interglacials[4].

The large sediment volumes building these fans are dominated by two sediment types accumulated during very short time periods: The first type are fans characterized by rapidly deposited glacigenic debris flows (GDFs), which indicate Pleistocene periods when eroding ice-streams reached the shelf edge and released the eroded sediment to the upper slopes (Fig. 1a)[1,5–11]. During shelf-edge glaciations, rapidly deposited glacial sediments are thought to be temporarily stored on the upper slopes, and eventually become unstable and generate GDFs with maximum runouts of >250 km[5,6,12,13]. These GDFs have been studied using 2D seismic data, which led to the conclusion that they have a lens-shaped geometry in profile view and a lobe-shaped expression in planar view[14,15]. GDF deposits are further documented to have transparent, generally incoherent acoustic facies with convex tops and pinch-out edges[13]. A temporary sediment storage at the upper slope with upcoming failure every 34–170 years during the last glacial maximum, and wedges and scars yet to be identified, was suggested for the Bear Island Trough Mouth Fan[5,12,16]. Sediment remobilization in the form of GDFs was further suggested as a relatively slow and non-disintegrating sediment transport process in very low-viscosity debris flows[6,13]. The deposits of these flows are poorly-sorted, matrix-supported diamicts with a sand content of up to 40% and higher shear strengths than glacimarine sediments[6,9,17]. Sediment cores showed that GDFs along Arctic margins have a finer grain-size composition than their Antarctic counterparts[3].

The second type are meltwater-dominated fans, which previously have been documented on mid-latitude, glacier-influenced margins (Fig. 1b). Large-volume meltwater delivery forms hyperpycnal flows, which result in the deposition of turbiditic sequences[2,18–21]. Turbidites detected on glacial fans are thus used as a proxy for meltwater delivery[13,22]. The relative importance of meltwater appears greater at lower than at higher latitudes[23]. Additional to the turbidite flows, deglacial plumites are released as the increased meltwater generated during the ice-sheet decay generates sediment plumes that also deposit with high sedimentation rates on the upper slopes[2,9,14,24–30].

Due to glacial erosion and the lack of distinct imprints of ice sheets on the paleo-shelves, trough mouth fan deposits are especially important when reconstructing pre-Weichselian glaciations and understanding glacial-interglacial cycles[31]. Sedimentological characterization of the uppermost meters of GDF and turbidite deposits are well established for a large variety of trough mouth fans[2,3,11,13]. However, ice-stream-dominated marine sedimentary systems are lacking extensive, high-resolution data. Depositional processes are thus still relatively poorly understood. The relevance of these marine depositional systems is growing due to economic activities on the seabed and its subsurface, especially in the Arctic region. Similarly, paleo-environmental and paleo-climatic reconstructions are gaining interest, specifically during past episodes of climatic transitions. Three-dimensional (3D) seismic reflection data offer new insights into the geometries, internal architecture and flow mechanisms of sediment remobilization processes[31,32]. Here, we test if trough mouth fan models derived from 2D seismic data are applicable in a 3D framework. This study aims to understand the nature of sediment delivery across the North Sea Fan during the last glaciation in three dimensions, to relate the sediments to the glacial history of the NE Atlantic margins (Fig. 2), and to discuss the implications for the formation of trough mouth fans.

The southeast–northwest-oriented North Sea Fan covers an area of c. 110,000 km$^2$ extending from water depths of 300 m at the shelf break into depths of 3500 m in the Norwegian Sea (Fig. 2a). Compared to other trough mouth fans, the North Sea Fan is a clear outlier due to its large fan area[3]. Sediment transport related to GDFs may have been operating on the North Sea Fan for the last 1.1 Myr, i.e., since the first ice-stream evidence in the Norwegian Channel[33]. The North Sea Fan received terrigenous sediment from hinterland-to-deep-sea sediment-routing systems with a catchment of c. 215,000 km$^2$, and comprises a sediment volume of c. 32,000 km$^3$[34,37]. Vertical erosion has been estimated to 164 m based on volume backstripping to the catchment[34], and has been modelled from 200 to 600 m for the route of the Norwegian Channel Ice Stream[35]. With a total annual output of 1.1 Gt of sediment (equivalent to 8000 m$^3$/yr per meter width of ice stream front), the Norwegian Channel Ice Stream was an extremely powerful sediment transport agent in the Late Quaternary[36,37]. Rapid sediment deposition associated with active ice streams resulted in the initiation of GDFs along a gently-dipping seabed (<1°)[6], which are deposited within massive clinoform packages consisting of low-amplitude seismic reflections (Fig. 2a). Nygård et al.[7] suggested six units of GDFs deposited in the Late Quaternary, and some of these thick units have been remobilized by megaslides[38]. Becker et al.[39] documented several sedimentation pulses characterized by coarse-grained sediment input to the Atlantic Margin during the last glaciation (Fig. 2b). The grounding line of the Norwegian Channel Ice Stream started to retreat from the continental shelf edge by c. 19 ka with an average retreat rate of 450 m/a and the channel was completely deglaciated by c. 17.5 ka[40].

## Results and discussion
**Seismic stratigraphy.** The sediment package related to the last glaciation (Weichselian) is defined by a continuous positive-amplitude

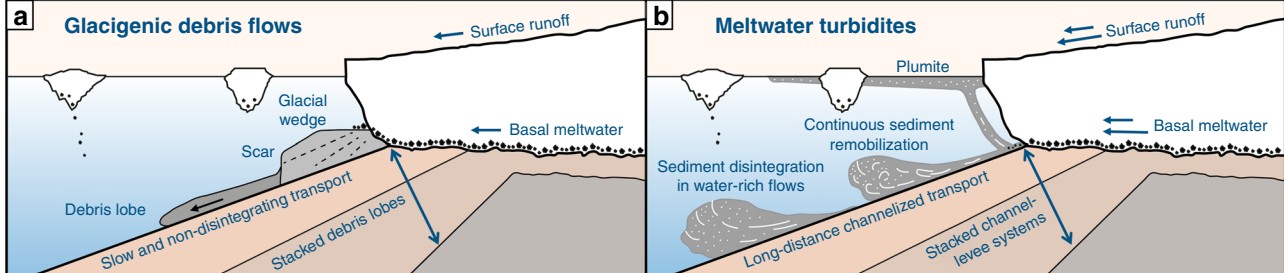

**Fig. 1 Two types of sedimentary systems forming trough mouth fans. a** Glacigenic debris flow dominated model implying temporarily stored sediment (glacial wedge) and non-disintegrating sediment transport. **b** Meltwater dominated model implying continuous channelized sediment transport in water-rich flows and surface plumites. Arrows indicating glacial meltwater are conceptual and not absolute values.

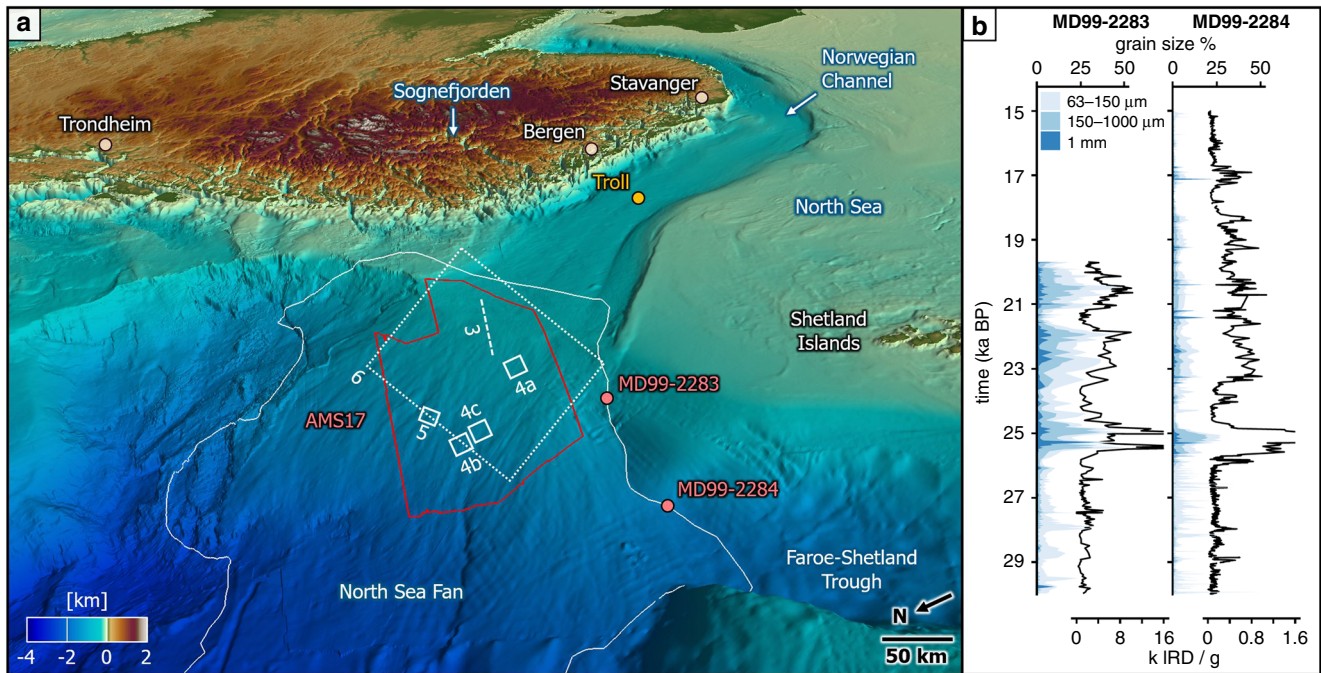

**Fig. 2 Oblique view of the bathymetry of the North Sea Fan. a** The extent of the high-resolution processed 3D seismic data (red line) and the North Sea Fan (white line) are outlined. Locations of the piston cores (red dots), Troll 8903 borehole (yellow dot), seismic profile of Fig. 3 (dashed line) and maps of Figs. 4–6 (white boxes) are shown. Vertical exaggeration is 25× for offshore and 2.5× for onshore domains. Scale bar approximate for central part of figure. **b** Piston cores used for age correlation (modified after[39]). Peaks in sand content (grain size >63 μm) indicate iceberg disintegration. IRD ice-rafted debris counts in thousands.

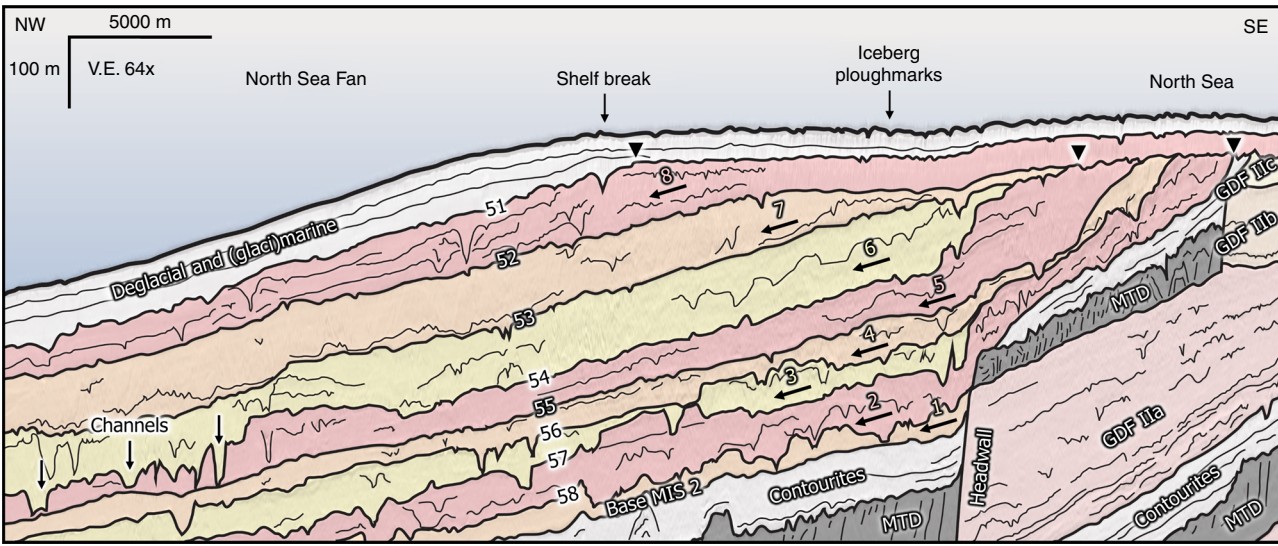

**Fig. 3 Seismic stratigraphy of the deposits related to the last glaciation (Weichselian, MIS 2) of the North Sea Fan.** Eight glacial sub-units are colored in yellow to red, and are indicated by arrows. The top reflections of the sub-units can have a negative-amplitude reflection (54, 56–58), or a positive amplitude reflection (52, 53, and 55). Deep, V-shaped depressions are recognized both at the top of the sub-units and along reflections within the sub-units. Contourites (light grey), Tampen Slide MTD (dark grey), GDFs related to the Saalian glaciation (MIS 6, light yellow to light red), and paleo-shelf break positions (black triangles) are shown. MIS marine isotope stage, MTD mass transport deposit. Profile located in Fig. 2a. V.E. vertical exaggeration. For uninterpreted seismic profile see Supplementary Fig. 2.

reflection at the base (Horizon Base MIS 2) and a continuous negative-amplitude reflection at the top (Horizon 51) (Fig. 3). The up to 450 m thick sediment package outcrops at the seabed of the deeper slopes, and is overlain by weakly layered deglacial and (glaci)marine sediments, up to 70 m toward the shelf break and c. 20 m on the shelf. For the 16,000 km² of this study, the sediment package

comprises a volume of c. 6400 km³. The homogenous seismic facies of the sediment package is interrupted by seven continuous high-amplitude reflections, separating eight sub-units with thicknesses of 20–80 m (Fig. 3). Four intercalating horizons have a negative-amplitude reflection (54, 56–58), and three horizons (52, 53, 55) have a positive-amplitude reflection (Supplementary Fig. 2). Horizons 51

and 54 can be traced into the North Sea, where they are characterized by lower seismic amplitudes.

The sediment packages related to pre-Weichselian glaciations (e.g., GDF II, Saalian) have a lower seismic amplitude response than the sediment sequence related to the last glaciation. GDFs of marine isotope stage 6 (GDF II, Fig. 3) have failed during the Tampen Slide[7,38], whose mass transport deposits can still be recognized as a package of deformed sediments onlapping a striking headwall (Fig. 3). The sedimentation of the glacial sediments is associated with a total paleo-shelf break migration of 5 km for the first five sub-units, and 16 km for the last three sub-units. The slope gradients of the paleo-seabeds have been reduced from 1.9° for the first five sub-units to 0.6° for the last three sub-units. Channels crosscut both reflections defining the borders of the sub-units and the reflections within the sub-units.

**Seismic geomorphology.** Three-dimensional seismic data have given rise to the discipline of seismic geomorphology, which is described by Posamentier et al.[41] as "the application of analytical techniques pertaining to the study of landforms and to the analysis of ancient, buried geomorphological surfaces as imaged by 3D seismic data". The mapped horizons of this study reveal multiple sharp, 5–50 m deep and 100–1000 m wide landform systems (Figs. 4 and 5), which locally truncate underlying reflections (Fig. 4a, b). The southeast–northwest oriented landform systems of the North Sea Fan are characterized by the highest seismic amplitudes, and can be traced from the shelf break to the deeper slopes over distances >150 km (Fig. 6, Supplementary Fig. 2). We interpret these landforms as channels, as the morphologies can be traced over large distances along the gently dipping seabed with typical channel geometries. The high-amplitude seismic response of the channels is characteristic for

channel infill (Fig. 6). The structure maps show flat terrains between the channels and local wedge-shaped deposits on the channel banks (Figs. 4 and 5a). Seismic attribute maps, however, show well-developed low-amplitude bars between the channels (Figs. 4c and 5). These bars have elongated geometries and a homogenous seismic facies (Fig. 5b). There is no correlation between seafloor relief (e.g., bathymetric lows) and channel occurrence on the evenly dipping paleo-seabeds of the North Sea Fan (Fig. 6). The channels on the North Sea Fan occur with a lateral spacing of 1–20 km, diverge and converge within short distances, and are characterized by a rather low than pronounced sinuosity (Fig. 4).

Channels with similar dimensions are identified along the seabed and in the subsurface in trough mouth fans of both hemispheres[8,42]. Glacial gullies, in contrast, are mainly expressed on the upper slopes of fans and have a rather straight expression and V-shaped incisions[2]. We interpret the reflections characterizing the elongated bars neighboring the channels as submarine levees of two types: a first type are asymmetric, wedge-shaped levees flanking submarine channels, similar to what has been described by Deptuck and Sylvester[43] for river-fed submarine fans (Fig. 5). The second, and more common type, are flat-topped levees deposited as uniform blankets on the pre-existing topography (Fig. 4). This levee geometry is different to levee geometries from fluvially derived systems, and has been observed on levees characterizing mud-rich turbidite systems[20,44]. The seismic character of the levee deposits is similar to the weakly stratified and transparent channel-levees described from the Northwest Atlantic Mid-Ocean Channel[45].

The high-amplitude reflections defining the channels (Fig. 4c) represent a strong impedance contrast to the underlying homogenous reflections. These contrasts in density and/or velocity most likely indicate coarser-grained sediment in active

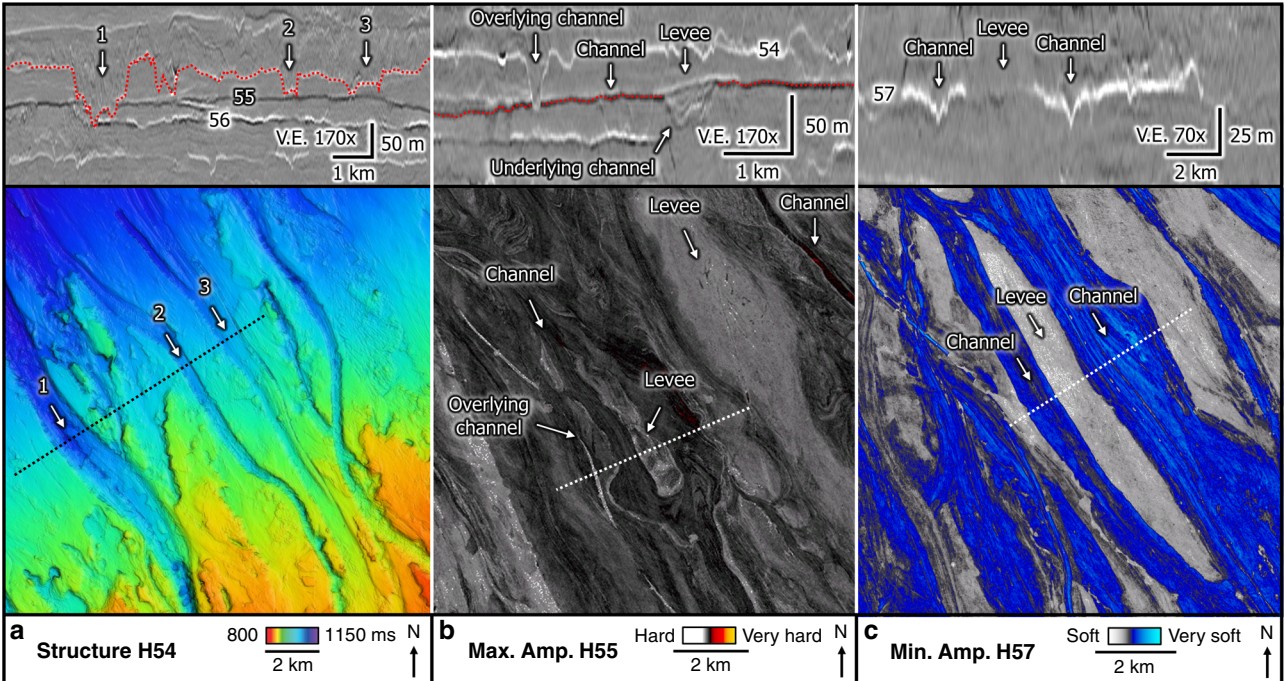

**Fig. 4 Examples showing the detailed morphology of channels, using zoomed seismic profiles, structure, and horizon attribute maps. a** Deeply eroded, wide channels on Horizon 54 (red line). Channel 1 crosscuts underling Horizons 55 and 56. **b** Channel-levee system on Horizon 55 (red line). The overlying channel (Horizon 54) is eroding into Horizon 55. Channels have harder amplitudes than levees. **c** Link between channels (infill) and levees on Horizon 57. The channels have high negative amplitudes (very soft), whereas the elongated levees have low negative amplitudes (soft). The location of the maps is shown in Fig. 2a and the stratigraphical position of the different horizons in Fig. 3. V.E. vertical exaggeration. For detailed interpretation of seismic profiles, see Supplementary Fig. 4. Seismic data courtesy of TGS.

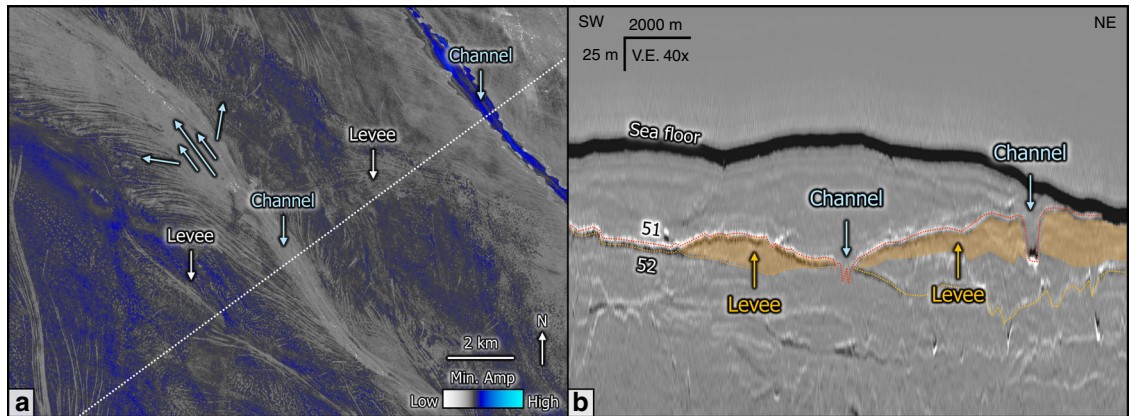

**Fig. 5 Seismic geomorphology of the uppermost channel-levee system. a** Minimum amplitude extraction of Horizon 51 showing seismic response of channel-levee system. **b** Seismic profile across channel-levee system highlighting levee geometry and levee facies. Horizon 51 (red line) and Horizon 52 (yellow line) are shown. Seismic data courtesy of TGS.

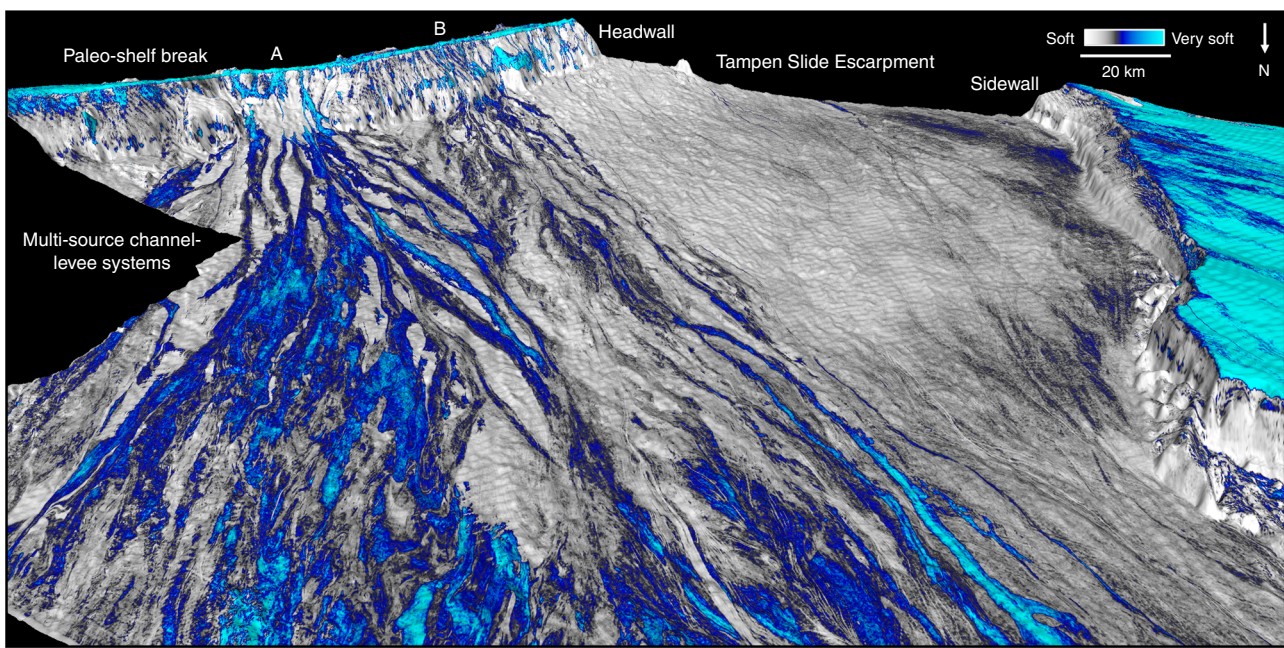

**Fig. 6 North Sea Fan at the beginning of the last glaciation.** 3D view of the Horizon Base MIS 2 (Fig. 2) draped by the minimum amplitude extraction in a window of 30 ms. The very soft bands (blue) are interpreted as channels of seismically distinct turbidite flows at the initiation of the last shelf-edge glaciation (t = 23 kyr). The Norwegian Channel Ice Stream, located at the shelf edge, forms two sediment sources at that time (indicated by A and B), from where meltwater turbidites fill the escarpment shaped by the Tampen Slide.

channels overlying fine-grained sediments. Similar conclusions were drawn on backscatter data on the Belgica Fan, where high backscatter returns from channel beds suggest a hard, eroded surface and/or a relatively coarse-grained component to the downslope flows that cut them[8]. Levees are characterized by low-amplitude reflections and lack the strong impedance contrast (Fig. 4c). Thus, levees rather indicate fine-grained sediment deposition originating from the suspensive load associated with hyperpycnal flows. Coarse-grained sediment commonly accumulates on the floors or at the mouths of submarine channels, whereas finer-grained sediment preferentially accumulates on channel banks and on adjacent aggradational levees[43,44,46]. Axial channel deposits have been documented to produce high-amplitude reflections in different fans globally[47]. In the case of the western Niger Delta slope[47], these reflections indicate predominantly sandy channel infill of turbiditic origin, whereas the levees consist of clay-grade sediment[48].

Based on detailed seismic interpretation, we conclude that the several 100-m-thick sediment sequence related to the last glaciation is dominated by channels and not, as previously suggested, by debris lobes. Extensive 3D seismic data are thus fundamental to correctly interpret glacial processes and its deposits.

**Implications on sedimentation and ice-stream activity.** The sediments of the last-glacial package of the North Sea Fan derive from subglacially transported sediments, which were deposited when the Norwegian Channel Ice Stream reached the shelf edge[6,36,37]. The eight sub-units within the uppermost sediment

package indicate that the ice stream oscillated eight times during the last glaciation (Fig. 3). Sediment has meanwhile continuously been transported downslope within commonly observed channel systems (Fig. 4), and fills the slide escarpment formed by the Tampen Slide (Fig. 6), resulting in a shelf-break migration of c. 5 km/kyr (Fig. 3). The data show that the ice stream delivered sediment from multiple sources and that the northern part of the fan was active first during the last glaciation (Fig. 6).

The growth and decay of the Norwegian Channel Ice Stream resulted in highly variable rates of sediment delivery to the continental margin[39]. The geometry and occurrence of channels identified at multiple levels within the sediment package related to the last glaciation (Fig. 3) document that the sediment delivery from the Norwegian Channel Ice Stream had a continuous pattern. The orientation of the channels does not significantly shift during the last glaciation (Fig. 6, Supplementary Fig. 3). Our data show a rather uniform sedimentation pattern within the same glaciation, while a shift of sediment depocenters has been observed for different glaciations on the North Sea Fan[49,50]. A relatively continuous subglacial release of material has previously been suggested as the origin for GDFs of this package[15]. Although straight or sinuous gullies related to the last glaciation have been described using side-scan sonar of the North Sea Fan[15], the seabed reflection of the 3D seismic data is rather mirroring deglacial than glacial processes (Figs. 2a and 3).

The Norwegian Channel Ice Stream was located at or close to the shelf break allowing dense sediment flows to develop from meltwater. The ice-stream-fed channels have a minimum age of c. 17.5 ka, as by that point the Norwegian Channel was completely deglaciated[40]. Most channels were formed before $18.7 \pm 0.2$ kyr, marking the retreat of the Norwegian Channel Ice Stream from the shelf edge[37,39]. The correlation of the five horizons with sediment cores from the distal part of the North Sea Fan indicate an age of 19–23 kyr for the sediment package related to the last glaciation[39] (Fig. 2b). Deposited within 4 kyrs, the up to 450 m thick package correlates with an average sedimentation rate of c. 100 m/kyr and a sediment flux of 1500 km³/kyr for the study area. Sedimentation rates in the most proximal core, just 2 km southwest of the sequence pinch-out, range from 0.5 to 1.5 m/kyr during the last glacial cycle[39]. The sedimentation rates within the North Sea Fan are thus 100 times higher compared to the rates outside the areas affected by direct ice-stream sedimentation. In line with previous studies (e.g., ref. [6]), the sediment supply associated with the channels outshines simultaneous glacimarine sedimentation. We further suggest higher deglacial sedimentation rates directly on the North Sea Fan, where the deglacial sediment package is up to 70 m thick (18 m/kyr), compared to cores from areas outside the fan (1 m/kyr). Similarly, Lucchi et al.[27] calculated extreme sedimentation rates of 34 m/kyr for the deglacial plumites from the upper-slope area of the Storfjorden Trough Mouth Fan in a period of less than 150 years.

**Meltwater turbidites**. The low-sinuous channels, whose downslope terminations expand over the extent of our data, indicate long-distance down-slope sediment bypassing. Long runout distances on low-gradient slopes were previously explained by excess pore fluid pressures[51] or the incorporation of a thin layer of ambient water underneath a subaqueous debris flow[52]. Muddy turbidites with long runout distances and feeding deep-sea fans have been documented in turbidite systems all over the world[53–56]. Based on the channel morphology and extent, we suggest sediment disintegration in water-rich flows as the dominating flow mechanism (Fig. 6). As channels and closely associated overbank deposits dominate the stratigraphy of the North Sea Fan, we suggest the fan to be maintained generally by glacial meltwater-sourced flows. We conclude that meltwater is an underestimated factor for the formation of trough mouth fans.

Sediment-carrying meltwater events can lead to the generation of erosive hyperpycnal flows[56,57]. Erosive sediment transport is shown by deep channels crosscutting underlying reflections (Fig. 4a, b). The flows have been more erosive at the uppermost slopes, where the channels are deepest. A lower degree of erosion observed by discontinuous channels in deeper waters of this study area could be supported by subaqueous turbidites running over antecedent turbidite deposits with no detectable remobilization, as shown for debris flows in experiments by Mohrig et al.[52].

Meltwater can transport large quantities of lithogenic particles derived from glacial erosion[58], and large turbidity currents are documented to have lost their freshwater after distances of up to 300 km[59]. An increased meltwater input from the Norwegian Channel Ice Stream could activate sediment-downslope transport in turbidity currents, and thereby explaining the long runout distances. The rapidly deposited sediment sequences along glaciated margins originally consist of poorly sorted and unstable glacial material (e.g.,[9,23]), but long-distance channelized sediment transport in turbidites might favor grain-size fractionation of these sediments[45]. The submarine channel network controls sediment distribution in the deep-water depositional system, and depending on transport distance and channel proximity, grain size might considerably vary along glacial reflections. Sediment cores collected from turbidite channels on the Squamish Delta contain multiple units of massive sands, with thicknesses of 1–2 m[60]. Such sand beds are resolvable by the seismic data used in this study, and high-amplitude reflections characterizing the channels can reflect sandy deposits. However, the acoustically transparent character of the sediment package between the high-amplitude reflections excludes the existence of thick sand beds, and indicates mainly silty and clayey deposits. A transparent seismic signature related to muddy material has been suggested for glacial fans in the Norwegian Channel[61].

**Model for the North Sea Fan during the last glaciation**. The presence of frequent well-defined stacked channel-levees in the proximal part of the North Sea Fan demonstrates that sediment has been transported downslope within commonly observed channel systems throughout the last glaciation (Fig. 4). We propose that the Norwegian Channel Ice Stream rapidly delivered eroded sediment (Fig. 6), resulting in multi-sourced turbidite systems on the fan. Turbidity currents linked to downslope flow of sediment-laden meltwater from the shelf edge could have directly formed the channels, which then functioned as conduits for focused turbidity current flow to the deep basin (Fig. 7)[11,45]. The bedload of such glacial turbidites can consist of medium sand and coarser sediments[45]. Additional to the turbidites, the sedimentation on the North Sea Fan was influenced by suspension settling from turbid-surface plumes released at the grounding line, which accumulated sediment in a clayey grain-size fraction (Fig. 7). Settling plume events are documented to trigger long-runout turbidity currents themselves on the Squamish Fan[62].

Different studies show that the period around the last glacial maximum was characterized by major input of meltwater events[22,26,63], and that trough mouth fans have highest growth in these periods[4]. Based on the glaciation history of the Norwegian Channel[39,40,64], we suggest that the high-density, sand-rich turbidity currents originate from the Norwegian Channel Ice Stream in the time period between 23 and 19 ka. These turbidity currents occurred during major meltwater discharges at the beginning of slope sedimentation. The last glacial maximum period is thus characterized by major input of meltwater events. As the turbidites are related to massive

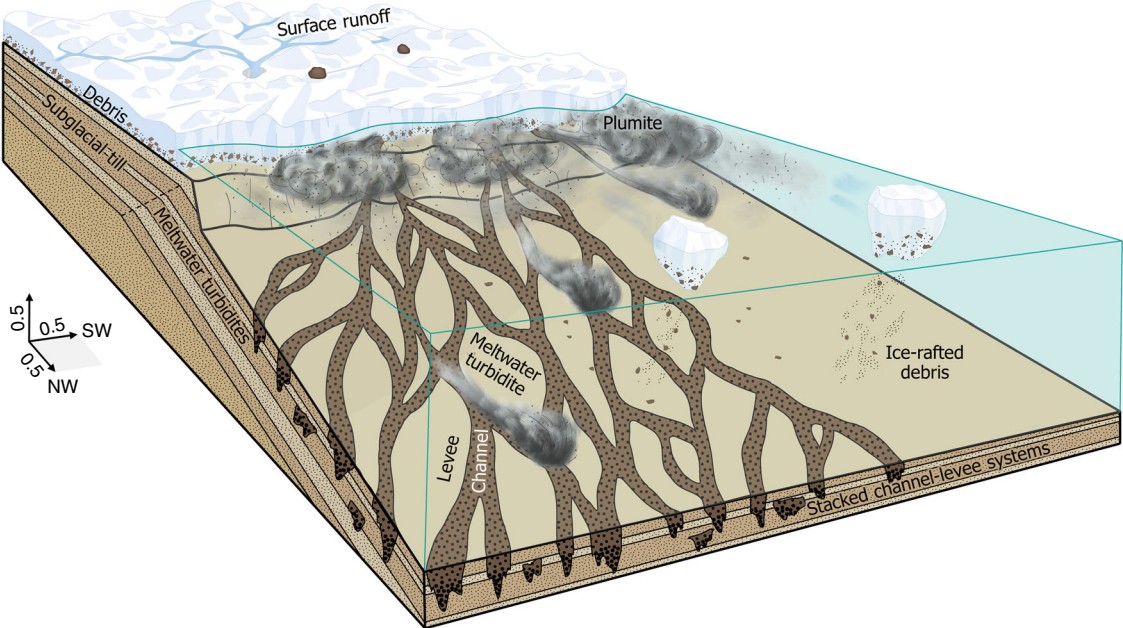

**Fig. 7 Conceptual model for sedimentation during the last glaciation (MIS2).** Meltwater turbidites and turbid-surface plumites are the dominating processes, and result in channel-levee systems on the North Sea Fan. Continuous sediment transport in water-rich flows build an up to 450 m thick sediment sequence in the time period of 23 to 19 ka. Dimensions are approximate, and given in km.

meltwater delivery, the turbidite intervals could correspond to short warmer periods, and a Norwegian Channel Ice Stream undergoing several smaller collapses within or after the last glacial maximum. Freshwater could largely be provided by seasonal meltwater discharge and iceberg calving, processes previously suggested by sedimentary records in other glaciated margins[65].

The new model suggests turbid glacial meltwater driven channelized sediment supply to be the dominating process shaping the mid-latitude North Sea Fan, and the Norwegian Channel to have acted as a major outlet for meltwater (Fig. 7). We suggest that coarse-grained turbidites at the beginning of a shelf-edge glaciation (Fig. 6), and delivery of coarser-grained material during a glaciation (Fig. 3), are the causes for the observed high-amplitude reflections. The data show that rapid glacial sedimentation is a continuous process during glaciations, with sediment accumulation two magnitudes higher in areas affected by channels. Voluminous meltwater production in periods with active ice streams at or close to the shelf break could increase turbidity current activity on the fan. Long-distance channelized sediment transport along gently dipping seabeds could explain kilometer thick glacial sequences hundreds of kilometers away from the shelf break.

**Implications for trough mouth fans.** The use of high-quality 3D seismic data allowed new interpretations of the style and variation of ice-proximal sedimentation on the North Sea Fan. Our study shows that sedimentation related to glacial meltwater played a fundamental role in the construction of the mid-latitude North Sea Fan compared to previously suggested sediment deposition by GDFs[6,7]. The stratigraphy of the mid-latitude North Sea Fan records a strong meltwater signal for the last glaciation, which distinguishes the fan from high-latitude fans with predominating low-water-content GDF deposition[5]. A strong meltwater delivery has also been suggested for mid-latitude depocenters of the Laurentian Fan and the Disko Fan[2,20], and for deglaciation in the high-latitude Storfjorden Fan[27]. The timing of the increased

meltwater discharge on the North Sea Fan correlates with turbidite deposits observed in both the Notre Dame and the Hawke Fans, which are associated to periods of major meltwater supply from 29 to 17 ka[22]. The volume and abundance of subglacial meltwater is largely controlled by strain heating and the geothermal heat flux beneath an ice sheet[66,67]. Freshwater fluxes are challenging to quantify and not necessarily correlated with fluxes of iceberg rafted debris[65]. Sediment core analysis of Becker et al.[39] proves that pulses of iceberg rafted debris not exclusively occurred between 23 and 19 ka, and we conclude that meltwater and iceberg supply to the NW Atlantic are asynchronous processes. A strong meltwater signal suggested by the glacial turbidite systems indicates that the North Sea Fan was probably a warmer environment during full-glacial and deglacial conditions compared with the more northerly glacial depocenters.

Gales et al.[3] suggested that ice-sheet drainage basin size influences the abundance and volume of subglacial meltwater released from beneath an ice sheet, and that turbidity current activity would increase in areas of greater meltwater. Turbiditic sedimentation is dominating the North Sea Fan, which is characterized by a large drainage basin. Similar process could dominate fan evolution of the Prydz Channel Fan and the Crary Fan in Antarctica, which both have ice-stream drainage basin areas of >1,000,000 km². High-resolution 2D seismic profiles and sediment cores often only cover the very uppermost meters on trough mouth fans[2,3]. The 3D seismic data of the North Sea Fan evidence that sedimentation of the complete last-glacial package took place through an overbank relationship to the channels. Glacial turbidites as a dominating trough mouth fan process have been documented for thinner sediment packages from other glaciated margins (e.g.,[2]), but never for a thickness comparable to the late glacial sequence of the North Sea Fan. In contrast to other settings[2,11,68], we do not observe any association between downslope process and slope gradient. A strong meltwater contribution could further imply that the Norwegian Channel Ice Stream had not to be positioned exactly at the shelf break during the deposition of the thick sediment package. The absence of shelf-edge glaciations is thus not

excluding high sediment accumulation rates. Glacial sediment could instead be delivered by subglacial meltwater from an ice margin that was no longer at the shelf edge, as suggested for the Donegal Barra Fan[24,69].

The detailed interpretation of the 3D seismic data shows that high-energetic hyperpycnites deposit up to 450 m of glacial turbidites, whereas low-water content GDFs could not form. Implied sedimentation rates of 100 m/ka outpace turbidite sedimentation rates of 1–3 m/ka in other fans[22]. Large volumes of rhythmic turbidites along glaciated margins are partly related to subglacial outbursts (e.g.,[20]). Therefore, the North Sea Fan might record multiple large outburst events during the last glacial maximum. The data further indicate that rapid meltwater-driven sedimentation dominate all of the last glacial sequence. Such a sedimentation pattern is in contrast to observations from other ice sheets, where sedimentation is changing from low-energy at the beginning to increased discharges at a later stage of the glacial cycle[13].

The extensive 3D seismic data set presented here allows better assessments of the significance of meltwater pulses during glaciations, and is thus relevant to more strongly constrain glacial and deglacial ice-sheet evolution. Strong sediment-laden turbidity current systems dominating glacial sedimentation are applicable for glacial settings with potential of high meltwater delivery and catchments with sediment available for erosion. Differences in the mode of sediment delivery to the continental slope and deep-sea basin strongly affected the evolution of the North Sea Fan. These differences most likely result in a distinct morphology of mid-latitude fans and their high-latitude counterparts.

## Methods

**Seismic data**. The study is based on seismic interpretation of 16,000 km$^2$ of high-resolution, industry-standard processed 3D seismic reflection data collected in 2017 from the proximal North Sea Fan (Fig. 2a). The data were collected by TGS using a triple-sourced airgun with a volume of 3000 in$^3$ and a shot point interval of 12.5 m. The acquisition consisted of twelve 8100 m long streamers, which were separated by 112.5 m. A high-resolution volume at 2 ms sample rate and 6.25 × 18.75 m binning was designed to increase the resolution of the shallow stratigraphy. The data for this volume have been cut at minimum of twice seabed time and 5000 ms two-way time before zero-phasing. The 3D seismic reflection data allow to image the buried sediment packages in a resolution of 2 m vertically and in a bin size of 20 × 5 m horizontally.

**Seismic interpretation**. Six seismic horizons within the last glacial sediment package were picked with an in-line spacing of 150 m, followed by gridding, horizon attribute extraction, sediment volume calculations, and seismic geomorphological interpretation. Seismic attributes, such as the minimum and maximum amplitudes of prominent reflections, provide additional geological information that cannot be extracted from structure maps and allow an improved geological process interpretation. The interpreted surfaces are characterized by both hard and soft reflections (Supplementary Fig. 1). P-wave velocities of 1500 and 1700 m/s were used for time-to-depth conversion of the water column and the last glacial sediment package, respectively[7]. We merged interpretations of regional 2D and 3D seismic data with bathymetric data from GEBCO 2014 to image the bathymetry of the study area (Fig. 2a).

**Chronostratigraphy**. This study focuses on the uppermost 500 m below the seabed, and follows the chronostratigraphy previously established for this sediment package, which is dominated by GDFs related to the last glacial cycle[7]. Glacial chronologies previously established by piston cores next to the North Sea Fan cover the last glacial cycle[39], and core ties allow constraining the ages of the sediment package of this study (Fig. 2b).

## Data availability

The 3D multiclient seismic data are part of the Atlantic Margins multiyear program that covered more than 50,000 km$^2$ from 2017 to 2019 and were acquired with triple source and 12 streamers. This data was provided courtesy of TGS and is not publicly accessible. Horizons and shapefiles are available upon reasonable request to the corresponding author.

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

## Acknowledgements

We thank TGS for allowing to publish the seismic profiles and seismic images. S.P. acknowledges the support from the Research Council of Norway (RCN) through its Centres of Excellence funding scheme, project 22325, as well as the ARCEx partners and RCN (grant number 228107).

## Author contributions

B.B. is the main author of the paper and led the analysis and interpretation of the seismic data, produced the figures with input from all authors, developed the core concepts of this study and wrote the manuscript. S.P. helped in seismic data analysis and significantly contributed toward the development of the core concepts of this study. L.W.M.B. contributed to the interpretation on ice-sheet dynamics and sedimentary deposits. R.M. was involved in data collection and provided access to the 3D seismic data. All authors contributed to writing the paper.

## Competing interests

The authors declare no competing interests.
