## [Peer Review File · Nature Communications]

Reviewers' comments:

Reviewer #1 (Remarks to the Author):

The authors have a high quality and very interesting dataset from the North Sea Trough Mouth Fan (TMF) off southern Norway. That dataset demonstrates the presence of a somewhat anastomosing channel pattern extending down-fan from the inferred ice margin near the shelf break. They use this channel pattern to infer that subglacial meltwater has played an important role in the evolution of the fan. I believe this is the first 3D seismic of such resolution and extent on a trough mouth fan, and the data and thus findings are superior to the previous study by Waage et al. (2018) on the Bear Island TMF. They also argue that the older hypothesis that glacial debris flows (GDFs) on the North Sea TMF were initiated by episodic failure of grounding line till wedges is not supported by their seismic imagery. All these exciting new findings certainly deserve publication.

However, I cannot agree with the interpretation of the data as presented. It may be that my interpretation is not supported by the authors' data and the critical evidence needs to be more clearly presented. The essential hypothesis of the authors is that the deposits on the TMF are rapidly accumulating "levee" deposits. The images illustrated show no clear evidence that deposition took place through an overbank relationship to the channels. In more distal turbidites, we look for wedging (thinning) of strata away from the channel as the best evidence of a depositional levee.

The alternative hypothesis for their observations is that TMFs grow by three main processes. (1) One is quasi-continuous supply of water-rich basal debris in the ice stream as a GDF. I don't know the North Sea Fan in any detail, but GDFs have been cored and imaged in high-resolution 2-D seismic on a number of TMFs. On Trinity TMF on the eastern Canadian margin (Tripsanas and Piper, 2008, *J. sed. Res.*) wedge like terminations have been imaged with high-resolution sparker and cored. (2) A second process, when more water is available, is the cutting of erosional channels by hyperpycnal flow. The authors show this process at 8 or 9 horizons on the North Sea Fan. On Trinity TMF, we recognised three such developments of erosional channels. (3) On the eastern Canadian margin, mud turbidites appear to be a very important high sedimentation rate facies on some TMFs. Not written up very well, but see Roger et al. 2013 (*Can J Earth Sci*), Saint-Ange and Piper, section 2.7 in Stokes et al. 2015 (*QSR*) and Figs. 5-7 of distal deposits in Rashid et al. (2019, *QSR*). It is not clear that these are "levee" deposits in the normal sense of the word. They deposit as a uniform blanket on pre-existing topography (e.g. recent Leng et al. papers on Laurentian Fan, Piper et al. *Palaeo3* 2007). They may initiate from muds in suspension in subglacial meltwater by a range of processes originally well discussed by Hesse and with recent analogues in papers on Squamish Delta by Hizzett et al. and Hage et al. (4) When I read the abstract, I wondered if I was going to find a good discussion of watery debris flows that are well known on land and produce constructional levees. The only reference I can pick off the top of my head for a marine example is Longva et al. (2008) *Mar. Geol.*, but I am sure there are others. So given my knowledge of other TMFs, I cannot accept the authors' hypothesis that channelized meltwater is the dominant process producing GDFs on all TMFs. In particular, it seems sedimentologically inconsistent with known cored GDFs for example on Bear Island TMF and Trinity TMF. I can accept that in particular at lower latitudes with lots of meltwater that erosional meltwater channels play an important role and that thick rapidly accumulating mud "turbidites" (*sensu lato*) may be the dominant constructional element of a TMF and may be preserved in interchannel areas and thus regarded as levees. But these have a very different seismic signature in high-resolution 2-D seismic from classic GDFs and are quite different in core.

Part of the problem here is that historically (King, Nygard) seismically amorphous units on the North Sea fan have been interpreted as GDFs, but this interpretation might be wrong. A GDF is a well-established sedimentological term for a depositional facies that has been cored and imaged with high resolution seismic. If the interpretation of seismically incoherent units on the North Sea Fan is wrong (quite possible, based on my knowledge of Labrador Sea TMFs, see Deptuck et al. 2007, Roger et al. 2013) then this needs to be handled in a different way to avoid apparent circular reasoning. The paper is peppered with references to GDFs that are really just amorphous

seismic units at the resolution of the seismic system used.

There is not enough discussion and evaluation of the main hypothesis of the authors, yet the paper is scattered with a lot of material on erosion and accumulation rates that have been known for 2 decades. Perhaps the authors have better resolution on these rates, but no scientific consequences seem to arise from the more precise data. The paper seems curiously unfocused in this regard.

POINTS OF DETAIL

Title. OK, except it should be "some Trough Mouth Fan Formation"

Fig 1. I find it very difficult to visualise these perspective oblique views with such a variable scale (why less VE on land, it won't obscure anything). Can you at least add a 5 or 2 degree lat-long grid and put the scale in the middle of the image where it is correct. What does k before IRD mean?

65 rates measured in thickness/time, not just metres

68. Sentence Rapid ... is repetitive

110. I don't think this is a significantly new finding. If it is, explain why later.

115-122 Is this important. Is it used later.

193 do you mean 3a not 2a

195-196. Not demonstrated. No seismic tie is shown to the cores used for chronology. Does one exist. Put in Supplementary Data.

201-206 Nothing new here. Is it important?

209 levees not demonstrated. Could be erosional remnants. No demonstrated constructional feature.

220-231 May be true for low latitude TMFs but not all TMFs (see title) and different from GDFs. 167 underlying

168 label 55 and 56 on figure

176-183 not much new here

187-188 what is the line of reasoning here

Fig. 4. Different perspective from Fig. 1. Very confusing. Either show outline box on Fig. 1, or superimpose a lat-long grid.

There is the seed of an important paper here. But the authors need to take account of what seem to be differing definitions of both TMF and GDF and be less expansive in their claims of a globally applicable model.

David J.W. Piper

david.piper@canada.ca

Reviewer #2 (Remarks to the Author):

Unlike low-latitude, river-fed marine sedimentary system, for which an immense scientific literature is available, high-latitude, ice-stream-dominated marine sedimentary systems suffer lack of extensive, high resolution data, so that the depositional processes are still relatively poorly understood. The relevance of these marine depositional system is growing for economic activities on the seabed and its subsurface, especially in the Arctic region, and for paleo-environmental and paleo-climatic reconstructions during past episodes of climatic transitions.

I find the data presented in this paper outstanding, novel and very important to improve our understanding of the mechanisms of growth of trough-mouth fans. The paper is well written. Excellent illustrations display the data and the methods, are addressed in detail.

I only have 2 general comments regarding the role of meltwater on the sedimentation on TMFs and the interpretation of channels, that should be better addressed in various parts of the manuscript. I have included several comments in the attached document in MS_Word format, that I summarise below:

1) ROLE OF MELTWATER

The way sedimentation on TMFs is presented as a baseline for this paper is exclusively focussed on the glacial debris flows traditionally believed to be generated at the grounding line during glacial maxima. This sedimentation is what generates the Grounding zone wedges on the upper slope (and on the continental shelf during step-wise ice stream retreat).

However, recent studies have demonstrated that TMFs are built also by another type of high sedimentation rate sedimentary mechanism: de-glacial plumites, released as the increased meltwater generated during the ice-sheet decay generates sediment plumes also deposited with high sedimentation rate on the upper slope. Both sediment types are deposited rapidly during very short time periods. You refer to this only at the very end of the section IMPLICATIONS ON SEDIMENTATION AND ICE-STREAM ACTIVITY (line 202). I think the issue of melt-water induce sedimentation as plumites (so different mechanism from the one you are describing here) should be given proper emphasis throughout the paper.

Again on the role of meltwater, I find that there is not a proper discussion on the role of meltwater with respect to glacial stages. Your pile of 8 debris-flow deposits are interpreted as the sedimentary expression of MIS-2. This is based on sediment core results, mainly the work by Nygard. MIS2 is the last cold MIS, spanning ~24 to 12 ka. MIS-2 comprises the LGM and the following deglaciation. Because meltwater is produced largely by the decay of the ice sheet, wouldn't it be more reasonable to relate the melt-water related debris flow to the post-LGM decay of the ice sheet? I cannot see how the postulated massive melt-water production needed to deposit the ~400 m-thick deposit could originate during the LGS, when the climate is the coldest of the entire 100 ka cycle.

2) INTERPRETATION OF CHANNELS

The typical morphology of TMSs for sediment transport is that of gully-systems. These are generally confined to the upper slope. Small, V-shaped incisions interpreted as the product of meltwater events generate density (hyperpycnal) flows that do not evolve in proper turbidity flows.

Channels, as you say, do occur on TMFs, but they are a minor feature.

Before concluding that what you image in the seismics are channels instead of gullies, I suggest that you compare the two and then provide all the evidences that lead to your interpretation. It seems to me you go straight to the interpretation of channels without a proper critical analysis. The levees are surely an important diagnostic element. However, the geometry of levees, at least in fluviially derived systems, is a bit different. I suggest that you include at least one good seismic image of levees. In the comments provided I point out to one of your figures that is not really convincing about the evidence of channels and levees.

In summary with an improvement of the paper by addressing the two points above (plus other minor suggestions provided in the attached file) I think the manuscript could be easily become ready for publication. It surely deserves it.

Angelo Camerlenghi

Reviewer #3 (Remarks to the Author):

This contribution from the author proposes a new conceptual model for the formation of glacial debris flows that dominate trough-mouth fan stratigraphy based upon 3D seismic data collected across the North Sea Fan. Using this high quality data the authors indicate the importance of meltwater and long-distance transportation of water-rich flows in TMF formation, which they argue

is currently underestimated.

This is a well-written manuscript. The seismic data displayed, and the geomorphic detail shown is quite impressive. Stacked channel and inter-channel features are clearly visible and the detail achieved does allow for process interpretation. I think the author present enough data (although better ground truthing of the seismic data would improve this) to support their conceptual model. However, I do wonder to what extent that this is a completely new concept. I refer the authors to the recent publication of Cofaigh et al 2018 'The role of meltwater in high-latitude trough-mouth fan development: The Disko Trough-Mouth Fan, west Greenland.' How does the conceptual model presented here differ from the meltwater dominated TMF development put forward by Cofaigh et al? The authors also neglect to link with other Arctic (Gales et al, 2018) and mid latitude (e.g., Piper et al 2007 and 2016; Hesse et al 1997 and 1999) TMFs. Due to the lack of reference to other TMF formation, I was left wondering whether this conceptual model was for the North Sea Fan only, or for all TMFs. The wider context therefore needs to be more explicit and thus refer to other research mention above would be useful.

However, if this conceptual model is an alternative process for TMF development, is it too simplistic to propose a "one-size fits all" model to TMF. Gales et al (2018) paper indicates that there is a difference between Arctic and Antarctic TMFs sediments and therefore process. How does this model fit with this research and also other research that implies a more 'traditional' view of TMF development? Can you rule out other the models entirely? If so, why? I would argue that different processes building TMFs are possible and this is a new example. I would like to see a bit more depth in the manuscript related to this.

Linking to the above, the authors state in the abstract and elsewhere in the text that freshwater supply is '...an underestimated factor for sedimentary processes active during glacial-interglacial cycles.' I would argue that this is not necessarily the case (e.g., the examples mentioned above and other studies). Again these papers need to be mentioned in the text.

Finally, I not sure that the complete process and sequence of the North Sea Fan development during the last glacial is a clear in the current manuscript. For instance, lines 187-189. The first sentence states that the sediment delivery '...had a continuous pattern, rather than occurring in pulses every several decades.' In the next sentence you mention how the channel orientation doesn't shift '... during these eight pulses...'. This seems contradictory.

This is an impressive dataset and worthy of publication however, before this article can be accepted a thorough check through literature on TMF formation and sedimentary processes needs to occur. A clear statement of the wider context of this model is needed with reference to where it is/is not applicable. Finally, can you clarify how the sediment sequence was produced i.e. continuous or occurring in pulses. I am therefore suggested major revisions.

Minor comments:

1. It is the reviewers preference to hyphenate trough-mouth fan.
2. Weichselian, last glaciations, last glacial maximum, MIS2; Are all needed or can you jut pick one. The addition of a broad age period would be helpful.
3. Line 48-50: TMF sediment characteristics. The new Gales et al (2018) indicates that sediment characteristic can vary depending on Location e.g., Arctic vs Antarctic. Should this be so simplified?
4. Line 81: Coarse sand is >500 microns. >63 would be all sand including fine.
5. Line 87: Suggest rewriting 'separated by 112.5 m' to '...which were 112.5 m apart.'
6. Line 90: Bit awkward, suggest you rephrase
7. Line 91: I'm not sure what you mean regarding the resolution of 20x5 m horizontally. Is that 100m resolution or between 20 to 5 m resolution or an area 20 by 5 m?
8. Line 97: 1800 m/s seem quite a high p-wave velocity. What is the rational for this velocity? I

didn't see a p-wave velocity I Nygard et al (2005) although I could be mistaken.

9. Line 106: 'related to the last glaciation (Weichselian)...'. You have already use Weichselian (line 38) so the 'last glaciation' bit should go earlier.

10. Line 135. Personally, I'm not a fan of quoting, but if you do add a page number.

11. Line 146: suggest you change the end of the sentence to '...are characterized by low sinuosity.

12. Line 187-188. '...in pulses of several decades.' Does the sedimentation rate and the resolution of the seismic data allow you to state this?

13. Line 178: You mention 8 oscillations of the Norwegian Channel Ice Stream which is very interesting when thinking of ice-stream dynamics. I was just wondering whether other records show such an oscillating margin and whether you can say how far from the shelf edge is the ice margin retreated and advanced?

14. Line 189: I am probably misinterpreting this, but to me this line suggest that sediment supply is from the NW to the SE. I would have thought this was the other way round.

15. Line 191-193 is awkward and I suggest you rewrite. Also, you normally refer to the depressions as channels, here you use chutes. Stay consistent.

16. Line 202: I would avoid using 'outshines'.

17. Lines 229-230: I would argue there is a lot of evidence that meltwater derived sediments reach the shelf edge when the ice margin has stepped back. This is often in the form of plumites and turbidites e.g., the Donegal Barra Fan. How is your turbiditic debris flows different to these? Is the sediment size different?

18. Line 234: suggest you change the wording from 'vanishing' to discontinuous.

19. Lines 238-240: What are the channel features cutting into? Can you categorically say that debris lenses did not exist? Could lenses have been incised and reworked? Could preservation be an issue?

20. Line 241-242: You state the conceptual model 'implies', I don't think this is correct. A conceptual model is produced from your interpretation of the stratigraphy, therefore it will 'imply' what you design it to. I suggest you rephrase this sentence.

21. Figure 4: add a reference for the 'previous model'.

22. Figure 4: what are the double arrows representing?

23. Figure 4b: how is this different to the conceptual model in Cofaigh et al (2018)?

Reviewer 1 (David Piper)

- 1) The images illustrated show no clear evidence that deposition took place through an overbank relationship to the channels. In more distal turbidites, we look for wedging (thinning) of strata away from the channel as the best evidence of a depositional levee.**

We more clearly present critical evidence of the overbank relationship and channel-levee formation by adding Figure 5 to the manuscript. The levees in Figure 5 also show thinning of strata away from the channel. In lines 218-225, we explain the two types of levees identified in the seismic data.

- 2) The alternative hypothesis for their observations is that TMFs grow by three main processes. (1) One is quasi-continuous supply of water-rich basal debris in the ice stream as a GDF. I don't know the North Sea Fan in any detail, but GDFs have been cored and imaged in high-resolution 2-D seismic on a number of TMFs. On Trinity TMF on the eastern Canadian margin (Tripsanas and Piper, 2008, *J sed. Res.*) wedge like terminations have been imaged with high-resolution sparker and cored. (2) A second process, when more water is available, is the cutting of erosional channels by hyperpycnal flow. The authors show this process at 8 or 9 horizons on the North Sea Fan. On Trinity TMF, we recognised three such developments of erosional channels. (3) On the eastern Canadian margin, mud turbidites appear to be a very important high sedimentation rate facies on some TMFs. Not written up very well, but see Roger et al. 2013. (*Can J Earth Sci*), Saint-Ange and Piper, section 2.7 in Stokes et al.**

We appreciate the reviewer's comment and mention that GDFs have been cored in previous studies (lines 61-63; 67; 69-71; 91-93). We added a paragraph on the "Model for the North Sea Fan during the last glaciation (lines 383-424) and created a new figure to better illustrate the processes active during the last glaciation (Figure 7). Figure 7 highlights the glacial and sedimentological setting present to form the last-glacial sediment sequence. We conclude that a combination of meltwater turbidites and plumites were the dominating processes (lines 421-422; 391-393). We further conclude that the last-glacial sediment sequence is dominated by glacial channels rather than the previously suggested debris lobes (lines 243-245).

- 3) It is not clear that these are "levee" deposits in the normal sense of the word. They deposit as a uniform blanket on pre-existing topography (e.g. recent Leng et al. papers on Laurentian Fan, Piper et al. *Palaeo3* 2007). They may initiate from muds in suspension in subglacial meltwater by a range of processes originally well discussed by Hesse and with recent analogues in papers on Squamish Delta by Hizzett et al. and Hage et al.**

This concern was addressed in "Point 1". Two types of levees identified in the seismic data are described in lines 218-225, and according references were added to the manuscript.

- 4) When I read the abstract, I wondered if I was going to find a good discussion of watery debris flows that are well known on land and produce constructional levees. The only reference I can pick off the top of my head for a marine example is Longva et al. (2008) *Mar. Geol.*, but I am sure there are others.**

We better addressed the process of "meltwater turbidites" in lines 313-381 by improving the existing text. Longva et al. (2008) is referred to for the interpretation of the muddier sediment deposits, which most likely are related to the suspension cloud of the turbidites (lines 373-374).

- 5) **I cannot accept the authors' hypothesis that channelized meltwater is the dominant process producing GDFs on all TMFs. In particular, it seems sedimentologically inconsistent with known cored GDFs for example on Bear Island TMF and Trinity TMF. I can accept that in particular at lower latitudes with lots of meltwater that erosional meltwater channels play an important role and that thick rapidly accumulating mud "turbidites" (sensu lato) may be the dominant constructional element of a TMF and may be preserved in interchannel areas and thus regarded as levees. But these have a very different seismic signature in high-resolution 2-D seismic from classic GDFs and are quite different in core.**

We refer to the studies highlighting the cored GDFs (lines 61-63; 67; 69-71; 91-93). In the revised manuscript, we are less expansive in our claims of a globally applicable model and highlight that the results are mainly applicable for mid-latitude trough mouth fans (lines 1-2; 410-412; 428-430; 478-479). In the last chapter, we further address implications to other trough mouth fans (lines 426-479). The differences between mid- (e.g. North Sea Fan) and high-latitude trough mouth fans are mentioned in lines 76-77, 428-430, 431-434 and 476-479.

- 6) **Part of the problem here is that historically (King, Nygard) seismically amorphous units on the North Sea fan have been interpreted as GDFs, but this interpretation might be wrong. A GDF is a well-established sedimentological term for a depositional facies that has been cored and imaged with high resolution seismic. If the interpretation of seismically incoherent units on the North Sea Fan is wrong (quite possible, based on my knowledge of Labrador Sea TMFs, see Deptuck et al. 2007, Roger et al. 2013) then this needs to be handled in a different way to avoid apparent circular reasoning.**

We fully agree with this comment, and add a small paragraph on this problematic and our new interpretation (lines 243-245).

- 7) **The paper is peppered with references to GDFs that are really just amorphous seismic units at the resolution of the seismic system used.**

The references mentioned by all three reviewers, and some additional ones, have been added to the manuscript. Glacigenic debris flows are cited with references from glacial depocenters all over the globe.

- 8) **There is not enough discussion and evaluation of the main hypothesis of the authors, yet the paper is scattered with a lot of material on erosion and accumulation rates that have been known for 2 decades. Perhaps the authors have better resolution on these rates, but no scientific consequences seem to arise from the more precise data. The paper seems curiously unfocussed in this regard.**

We highlight that previous studies mainly used shallow cores and 2D seismic data to make conclusions about the sedimentary processes. Here, we use the most extensive 3D seismic data set to make conclusions on a sediment package as thick as 450 m and extending over more than 17,000 km². The quality of the 3D seismic volume is high in both vertical and lateral resolution. Thus, we could interpret channel-levee systems previously not imaged and make conclusion on the complete sediment package. We also highlight that the last-glacial sediment sequence has been suggested to consist of debris lobes, whereas our study shows that it is actually dominated by glacial channels.

Line edits

9) Title. OK, except it should be “some Trough Mouth Fan Formation”

We added the word “mid-latitude” to the title.

10) Fig 1. I find it very difficult to visualise these perspective oblique views with such a variable scale (why less VE on land, it won't obscure anything). Can you at least add a 5 or 2 degree lat-long grid and put the scale in the middle of the image where it is correct. What does k before IRD mean?

We disagree with this comment. In order to highlight the area of interest we have to use a higher exaggeration for the submarine domain. The more than 2000 m high mountains of the Norwegian mainland are not important for the study and take away the focus from the submarine environment. As we are dealing with 3D data, we really want to show their potential, and visualize the setting in three dimensions. Thus, it is to distractive to add lat-long information. We double-checked the lateral scale, and it is correct for the place it is located right now. “k” before IRD stands for “kilo”, or thousand. We added this information to the figure caption (line 141).

11) Rates measured in thickness/time, not just metres

We agree with this point. However, the information is taken from a previous paper, which does not give thickness/time itself. In line 121, we added the volume per year in the same way it has been done in the paper by Nygård et al. (2007).

12) 110 I don't think this is a significantly new finding. If it is, explain why later.

The section “Seismic Stratigraphy” (lines 166-193) first describes the results of our study. Detailed description is fundamental to follow our interpretation. The description of the reflections is crucial for all the manuscript.

13) 115-122 Is this important. Is it used later.

See comment on “point 12”. The topography shaped by the Tampen Slide is fundamental for the sedimentation during the last glaciation. Shelf break migration and slope gradient is important to discuss both the amount of sediments and the type of sediment remobilization. Both aspects have later been used to explain sedimentation on the North Sea Fan (lines 269-270; 309-311; 346-348; 457-458).

14) 195-196. Not demonstrated. No seismic tie is shown to the cores used for chronology. Does one exist. Put in Supplementary Data.

This is a good comment. However, there are no existing wells penetrating this sequence on the slope. A well in future would be very much appreciated to ground-truth results gained from 3D seismic data.

15) 201-206 Nothing new here. Is it important?

The higher deglacial sedimentation rates of the North Sea Fan are important to put it into perspective compared to other glacial depocenters. This information has been used for comparisons later in the manuscript (lines 300-304).

16) 209 levees not demonstrated. Could be erosional remnants. No demonstrated constructional feature.

See detailed comment on "Point 1" and "Point 3".

17) 220-231 May be true for low latitude TMFs but not all TMFs (see title) and different from GDFs.

See detailed comment on "Point 5".

18) 168 label 55 and 56 on figure

Horizons 55 and 56 have been labelled on the figure.

19) 176-183 not much new here

This is the introduction paragraph into the discussion, and thus include some existing facts. However, the channelized sediment process and the amount of shelf-break migration within one glacial cycle described in that paragraph are new.

20) 187-188 what is the line of reasoning here

The reasoning of that sentence was indeed a bit unclear, and we improved the sentence (lines 278-279).

21) Different perspective from Fig. 1. Very confusing. Either show outline box on Fig. 1, or superimpose a lat-long grid.

We disagree with this comment. The aim of Figure 1 (now Figure 2) was to introduce the setting/study area, whereas the aim of Figure 6 is to zoom and highlight the sedimentary processes active during the last glaciation. In order to display this in the best way in 3D, we slightly rotated it. We followed the recommendation of the reviewer and show the outline box of Figure 6 on Figure 2.

Reviewer 2 (Angelo Camerlenghi)

- 1) **The way sedimentation on TMFs is presented as a baseline for this paper is exclusively focussed on the glacial debris flows traditionally believed to be generated at the grounding line during glacial maxima. However, recent studies have demonstrated that TMFs are built also by another type of high sedimentation rate sedimentary mechanism: de-glacial plumites, released as the increased meltwater generated during the ice-sheet decay generates sediment plumes also deposited with high sedimentation rate on the upper slope. Both sediment types are deposited rapidly during very short time periods. You refer to this only at the very end of the section IMPLICATIONS ON SEDIMENTATION AND ICE-STREAM ACTIVITY (line 202). I think the issue of melt-water induce sedimentation as plumites (so different mechanism from the one you are describing here) should be given proper emphasis throughout the paper.**

This is a very much appreciated comment and following the reviewer's recommendations strengthened the manuscript a lot. We relate to deglacial plumites in the following way:

- A paragraph on meltwater-dominated fans in the "Introduction" chapter (lines 72-81)
- Splitting the previous Figure 4 into new Figure 1 and Figure 6. Figure 1 in the "Introduction" is giving a proper background to the different types of trough mouth fans
- Discussing deglacial plumites as a fundamental process throughout the manuscript (e.g. lines 302-304; 391-394; 421-422)
- Meltwater-related sedimentation in chapter "Meltwater Turbidites" (lines 313-374)

- 2) **Again on the role of meltwater, I find that there is not a proper discussion on the role of meltwater with respect to glacial stages. Your pile of 8 debris-flow deposits are interpreted as the sedimentary expression of MIS-2. This is based on sediment core results, mainly the work by Nygard. MIS2 is the last cold MIS, spanning ~24 to 12 ka. MIS-2 comprises the LGM and the following deglaciation. Because meltwater is produced largely by the decay of the ice sheet, wouldn't it be more reasonable to relate the melt-water related debris flow to the post-LGM decay of the ice sheet? I cannot see how the postulated massive melt-water production needed to deposit the ~400 m-thick deposit could originate during the LGS, when the climate is the coldest of the entire 100 ka cycle.**

Previous studies support that the period around the last glacial maximum was characterized by major input of meltwater events (Lekens et al., 2006; Haapaniemi et al., 2010; Roger et al., 2013), and that trough mouth fans have highest growth rates in these periods (Dowdeswell and Elverhøi, 2002). We better state that the sediment package has been deposited from 23 to 19 ka (lines 292-293; 395-400; 423-424). We thus think that oscillations of the Norwegian Channel Ice Stream, with smaller decays, could be responsible for the meltwater turbidites.

- 3) **The typical morphology of TMSs for sediment transport is that of gully-systems. These are generally confined to the upper slope. Small, V-shaped incisions interpreted as the product of meltwater events the generate density (hyperpycnal) flows that do not evolve in proper turbidity flows. Channels, as you say, do occur on TMFs, but they are a minor feature. Before concluding that what you image in the seismics are channels instead of gullies, I suggest that you compare the two and then provide all the evidences that lead to your interpretation. It seems to me you go straight to the interpretation of channels without a proper critical analysis.**

We more critically analyse the glacial channels (lines 205-207) and explain why the observed landforms are not gullies (lines 216-218).

- 4) **The levees are surely an important diagnostic element, However, the geometry of levees, at least in fluviially derived systems, is a bit different. I suggest that you include at least one good seismic image of levees. In the comments provided I point out to one of your figures that is not really convincing about the evidence of channels and levees.**

See detailed response in "Point 1, Reviewer 1": We more clearly present critical evidence of the overbank relationship and channel-levee formation by adding Figure 5 to the manuscript. The levees in Figure 5 also show thinning of strata away from the channel. In lines 218-225, we explain the two types of levees identified in the seismic data. The comment on the figure labelling has been dealt with, and the figure is more clearly labelled now.

Line edits

- 5) **Lines 28-29. The role of freshwater is well regognized during deglaciation. Your results are novel for sediment transport processes during MIS-2, that comprises the LGM and the following deglaciation. I suggest that you link the meltwater driving process to stadial-interstadial cyclicity rather that glacial cycles.**

We dealt with this comment by deleting the word "interglacial". Now, it is stated that freshwater supply is an underestimated factor for sedimentary processes active during glacial cycles (lines 34-37).

- 6) **Lines 51-52. This statement implies that your hypothesis is that the identification of these GDFs deposit you find on the North Sea Fan is attributable to the improved geophysical tool (3D vs 2D). I agree that the imaging provided by the HR 3D seismic is fundamental. However, could there be a 'geological' reason why these melt-water derived debris flows are present on the North Sea Fan and not perhaps in other TMFs?**

We discussed the implications of the findings of our study for other glacial depocenters in a new paragraph (lines 426-479). Better including the study of Gales et al. (2019), we especially highlight the role of the ice-sheet drainage basin size of the North Sea Fan (lines 115-116; 447-451). It is also stated that meltwater might mainly be important for mid-latitude fans, compared to their high-latitude counterparts (lines 410-412; 427-446).

- 7) **Line 138. Elsewhere in the text you define your survey area as covering the 'proximal North Sea Fan». I agree that the dimensions of the fan are huge, and therefore the 150 km length downslope covered by your data imaging the channels demonstrate the exceptional (compared to other smaller TMFs) distance run by the debris flows. However, that distance is about half, or less of the total length of the fan. The reflectivity of the channels seems to decrease in the deepest part of your data set, so one may infer that the channels do not continue until the base of the North Sea Fan slope, as you imply in your conclusions.**

In order to make solid conclusions on this point, new 3D seismic cubes would have to be collected in the very distal part of the North Sea Fan. The sentence so far only refers to "deeper slopes over distances

>150 km”, which is supported by our extensive dataset. For conclusions of the more distal North Sea Fan, we would speak about distances of >300 km.

- 8) **Lines 139-141. In text you refer to inter-channel bars, banks and levees. In figures you only mention levees. Please use a consistent terminology or explain the difference. As an example, in figure 3b (upper pane, cross section) you indicate with an arrow a levee that corresponds clearly to a filled channel deposit, and a channel one a flat area. Maybe labels are misplaced?. This is confusing.**

The expression bar is in the revised manuscript only used for descriptive reasons and has been changed to levee after interpretation. The comment on the figure labelling has been dealt with, and the figure is more clearly labelled now.

- 9) **Lines 139-141. Also, the ‘flat terrain’ between channels as you define it, does not correspond to the geometry of levees, that are typically asymmetric wedged sedimentary accumulations, thicker close to the channel. Consequently, levees are not flat topped, but sloping away from the channel. I suggest you show as an inset, or Supplementary figure, the best image of a typical levee deposit you can extract from your data set, to make your interpretation really convincing.**

See detailed response on “Point 4”.

- 10) **Lines 146-147. I think Late Weichselian is more appropriate, or LGM. Last Glaciation would be interpreted as the entire 100 k yr glacial cycle (Weichselian cycle).**

The sedimentation of the North Sea Fan is dominated by deposits from 23 to 19 ka. Consequently, we refer to this time period (lines 161-163).

- 11) **Line 185. Why should the pulses be decadal? You cannot resolve the chronology with such a detail (decadal versus continuous)**

We agree on this point and removed this part of the sentence.

- 12) **Line 187. I cannot see the progression of sediment accumulation from NW to SE in your images.**

This point is difficult to explain without showing multiple images of reflections of the North Sea Fan. What we meant is that the ice stream initially builds channel-levee systems in the NW of the Tampen Slide Escarpment, and later in the SE. However, this point does not add much to the manuscript, and we deleted that sub-sentence.

- 13) **Lines 215-216. You propose that the same mechanism applies to other TMFs. It is true that seismic data sets of comparable resolution in 3D are missing elsewhere. So, the unique information you are providing could apply to TMFs worldwide, but there may be other peculiarities of the Norwegian Channel that justify the different architecture: e.g. an exceptional meltwater discharge system given the lower latitude with respect to other known TMFs, or Antarctic TMFs.**

See detailed response on "Point 6".

- 14) I think one important point to address is why you have large quantities of meltwater during a cold stage (MIS-2), or perhaps the it is only the last, deglacial part of MIS-2 that produces these debris flows?**

See detailed response on "Point 2".

- 15) It is very evident from your images that the uppermost continental slope, just below the shelf break shows a steeper topographic gradient, suggesting a sedimentary wedge like every other TMF shows. The steeper slope implies a classical GDF origin, with stiff diamicton. If so, the debris flows you image appear to come later, so right after the glacial maximum, when with the beginning of the ice sheet decay produces massive meltwater at the shelf edge.**

The uppermost continental slope just below the shelf break is steeper, as it is the headwall shaped by the Tampen Slide. This is discussed in the manuscript (lines 177-181; 269-271) and shown in illustrations (Figures 3 and 6). The Tampen Slide removed glacial sediments of MIS6 (Nygård et al., 2005; Bellwald et al., 2019), and the headwall is the contact between glacial sediments of MIS2 downslope and glacial sediments of MIS6 upslope (see Figure 3).

- 16) MIS-2 spans 24 to 12 ka. It includes all the de-glaciation period that followed the LGM. Most of MIS two, therefore comprises deglaciation rather than LGM. The huge sedimentary package formed by meltwater-derived debris flows may reflect the deglaciation part of MIS-2 rather than the LGM part.**

For the dating and ice sheet configuration, we refer to the studies of Nygård et al. (2005), Becker et al. (2018), Morén et al. (2018) and Sejrup et al. (2016). Previous studies confirm that the period around the last glacial maximum was characterized by major input of meltwater events (Lekens et al., 2006; Haapaniemi et al., 2010; Roger et al., 2013), and that trough mouth fans have highest growth rates in these periods (Dowdeswell and Elverhøi, 2002). Plumites can be found in the up to 70 m-thick sediment sequence above the "last-glacial" sequence. As our newly created Figure 7 shows, plumites might have been active at the same time as the glacial turbidites and might have some contribution to the seismically transparent package between the key reflections of this study.

Reviewer 3

- 1) How does the conceptual model presented here differ from the meltwater dominated TMF development put forward by Cofaigh et al? The authors also neglect to link with other Arctic (Gales et al, 2018) and mid latitude (e.g., Piper et al 2007 and 2016; Hesse et al 1997 and 1999) TMFs. Due to the lack of reference to other TMF formation, I was left wondering whether this conceptual model was for the North Sea Fan only, or for all TMFs. The wider context therefore needs to be more explicit and thus refer to other research mention above would be useful.**

We agree that the manuscript was short on discussing other trough mouth fans and went through an intensive literature check. Most importantly added papers are Hesse et al. (1997a; 1997b; 1999), Knutz et al. (2002), Leng et al. (2018), Longva et al. (2008), Lucchi et al. (2015), Ó Cofaigh et al. (2018), Piper et al. (2007; 2016), Piper and Normark (2009), Rashid et al. (2019), Rebesco et al. (2013), Roger et al. (2013), Stokes et al. (2015), Tarlati (2018), and Tripsanas and Piper (2008).

We include the model from Ó Cofaigh et al. (2018) at several passages (lines 73-75; 77-81; 434-437), and highlight the main differences of our study compared to their work (lines 91-93; 452-453; 455-457; 457-458). The wider context of this work has been more explicitly discussed in the new chapter "Implications for Trough Mouth Fans" (lines 426-479). The work of Gales et al. (2019) has been better integrated in the study (lines 69-71; 115-116; 447-451; 452-553).

- 2) However, if this conceptual model is an alternative process for TMF development, is it too simplistic to propose a "one-size fits all" model to TMF. Gales et al (2018) paper indicates that there is a difference between Arctic and Antarctic TMFs sediments and therefore process. How does this model fit with this research and also other research that implies a more 'traditional' view of TMF development? Can you rule out other the models entirely? If so, why? I would argue that different processes building TMFs are possible and this is a new example. I would like to see a bit more depth in the manuscript related to this.**

This is a very valid critic, and we added a new chapter to highlight where (and where not) the outcomings of this work fit with other glacial depocenters (lines 426-479). We also added a sentence to the introduction about differences in grain-size composition (Gales et al., 2019; Lines 69-71).

- 3) Linking to the above, the authors state in the abstract and elsewhere in the text that freshwater supply is '...an underestimated factor for sedimentary processes active during glacial-interglacial cycles.' I would argue that this is not necessarily the case (e.g., the examples mentioned above and other studies). Again these papers need to be mentioned in the text.**

As mentioned in "Point 1", we did a detailed literature check, and included/discussed relevant papers from glaciated margins all over the globe. We better highlight in what way our study indicates that freshwater supply is an underestimated factor (lines 345-352; 359-362; 387-390; 400-405; 410-412; 416-417; 423-424; 444-446; 469-470).

- 4) Finally, I not sure that the complete process and sequence of the North Sea Fan development during the last glacial is a clear in the current manuscript. For instance, lines 187-189. The first sentence states that the sediment delivery'...had a continuous pattern, rather than occurring in pulses every several decades.' In the next sentence you mention how the channel orientation doesn't shift '... during these eight pulses...'. This seems contradictory.**

See response to "Point 8, Reviewer 2".

- 5) **This is an impressive dataset and worthy of publication however, before this article can be accepted a thorough check through literature on TMF formation and sedimentary processes needs to occur. A clear statement of the wider context of this model is needed with reference to where it is/is not applicable.**

This is a very valid critic, and we added a new chapter to highlight where (and where not) the outcomings of this work fit with other glacial depocenters (lines 426-479). As shown in "Point 1", a thorough check through literature on TMF formation has been done. Literature on sedimentary processes has been updated, and the most relevant studies are Hesse et al. (1997a; 1997b; 1999), Hizzett et al. (2018), Joughin et al. (2004), Leng et al. (2018), Longva et al. (2008), Lucchi et al. (2015), Ó Cofaigh et al. (2018), Pattyn (2010), Piper et al. (2007), Piper and Normark (2007), Roger et al. (2013) and Tripsanas and Piper (2008).

- 6) **Finally, can you clarify how the sediment sequence was produced i.e. continuous or occurring in pulses.**

The stacked sub-units of channel-levee deposits are deposited in the time period between 23 to 19 ka and indicate continuous sedimentation (lines 269-271; 275-279; 281-283; 414-415; 423-424). The sub-sentence with the pulses has been deleted, as mentioned in "Point 11, Reviewer 2".

Line edits

- 7) **Weichselian, last glaciations, last glacial maximum, MIS2; Are all needed or can you jut pick one. The addition of a broad age period would be helpful.**

We reduced the use of this expressions but think that for some discussion and clarification it is still required. The word "Weichselian" only occurs now in passages and sediment sequences where we compare with pre-Weichselian time periods.

- 8) **Line 48-50: TMF sediment characteristics. The new Gales et al (2018) indicates that sediment characteristic can vary depending on Location e.g., Arctic vs Antarctic. Should this be so simplified?**

The work of Gales et al. (2019) mainly focuses on shallow sediment cores and seafloor expression of a multitude of trough mouth fans. In the "Introduction", we now state than the sedimentology varies between Artic and Antarctic trough mouth fans (lines 69-71).

- 9) **Line 81: Coarse sand is >500 microns. >63 would be all sand including fine.**

This mistake has been fixed, and the sentence has been rewritten (lines 140-141).

- 10) **Line 87: Suggest rewriting 'separated by 112.5 m' to '...which were 112.5 m apart.'**

We did not rewrite this sentence, as talking of streamer separation is the common way of expression in marine geophysics.

- 11) Line 91: I'm not sure what you mean regarding the resolution of 20x5 m horizontally. Is that 100m resolution or between 20 to 5 m resolution or an area 20 by 5 m?**

We rewrote the sentence so that it will be better understandable (lines 150-151). What we mean is that the geophysical data have a lateral resolution defined by a bin size of 20 x 5 m.

- 12) Line 97: 1800 m/s seem quite a high p-wave velocity. What is the rationale for this velocity? I didn't see a p-wave velocity in Nygård et al (2005) although I could be mistaken.**

Nygård et al. (2005) deals with the velocities of the last glacial sediment package (Nygård et al., Table 1; Nygård et al., Chapter 5.1 Seismic correlation and chronological constraints). For MTDs, Nygård et al. use 1900 m/s as velocity, whereas a velocity of 1700 m/s is used for what they define as glacial debris flows. We thus use 1700 m/s for the glacial channels and last-glacial sediment package.

- 13) Line 106: 'related to the last glaciation (Weichselian)...'. You have already use Weichselian (line 38) so the 'last glaciation' bit should go earlier.**

We reduced the word "Weichselian" to passages and sediment sequences where we compare with pre-Weichselian time periods.

- 14) 187-188. '...in pulses of several decades.' Does the sedimentation rate and the resolution of the seismic data allow you to state this?**

See response "Point 11, Reviewer 2": We agree on this point, and removed this part of the sentence.

- 15) Line 178: You mention 8 oscillations of the Norwegian Channel Ice Stream which is very interesting when thinking of ice-stream dynamics. I was just wondering whether other records show such an oscillating margin and whether you can say how far from the shelf edge is the ice margin retreated and advanced?**

To our knowledge, there are no other studies documenting this amount of oscillations. We further think that conclusions about ice-margin retreats and readvances might be too speculative, and the focus of an independent study. However, we state in the manuscript that the Norwegian Channel Ice Stream might have undergone several smaller collapses during the last glacial maximum (lines 401-403).

- 16) Line 189: I am probably misinterpreting this, but to me this line suggest that sediment supply is from the NW to the SE. I would have thought this was the other way round.**

See response "Point 12, Reviewer 2": This point is difficult to explain without showing multiple images of reflections of the North Sea Fan. What we meant is that the ice stream initially builds channel-levee systems in the NW of the Tampen Slide Escarpment, and later in the SE. However, this point does not add much to the manuscript, and we deleted that sub-sentence.

- 17) Line 191-193 is awkward and I suggest you rewrite. Also, you normally refer to the depressions as channels, here you use chutes. Stay consistent.**

We used gullies for consistency reasons. This was also recommended by Reviewer 2.

- 18) Lines 229-230: I would argue there is a lot of evidence that meltwater derived sediments reach the shelf edge when the ice margin has stepped back. This is often in the form of plumites and turbidites e.g., the Donegal Barra Fan. How is your turbiditic debris flows different to these? Is the sediment size different?**

We added discussion on meltwater-derived sediments from stepped-back margins in the discussion (lines 461-463), using the studies of Knutz et al. (2002) and Tarlati (2018). Our study shows an up to 450 m-thick sediment package, for which we can prove the presence of channel systems in three dimensions. The plumite package (related to deglaciation) is overlying the package characterized by glacial turbidites (Figure 3; lines 169-171).

- 19) Line 234: suggest you change the wording from 'vanishing' to discontinuous.**

We followed the reviewer's suggestion and replaced the wording.

- 20) Lines 238-240: What are the channel features cutting into? Can you categorically say that debris lenses did not exist? Could lenses have been incised and reworked? Could preservation be an issue?**

We can only make conclusions on strata that show seismic response. The seismically transparent facies between the key reflections could consist of some debris lenses, however without any clear shape. Our study concludes that the reflections we are able to interpret with confidence all have a channel-levee geomorphology. The channels most likely eroded into pre-existing muddy fraction of turbidites (lines 355-358; 370-374).

- 21) Line 241-242: You state the conceptual model 'implies', I don't think this is correct. A conceptual model is produced from your interpretation of the stratigraphy, therefore it will 'imply' what you design it to. I suggest you rephrase this sentence.**

We replaced "implies" by "suggests".

- 22) Figure 4: add a reference for the 'previous model'. Figure 4: what are the double arrows representing?**

We split this figure into two separate figures, and the new Figure 1 deals with "Glacigenic debris flows" and "Meltwater turbidites" dominated trough mouth fans. In the figure caption of Figure 1, we state that the arrows indicate glacial meltwater input (line 87).

- 23) Figure 4b: how is this different to the conceptual model in Cofaigh et al (2018)?**

See detailed response "Point 1, Reviewer 3"

REVIEWERS' COMMENTS:

Reviewer #1 (Remarks to the Author):

I read the rebuttal letter and felt that the authors had done a good job. I read the manuscript and thought that this paper is enormously improved. But I still have some difficulty with some of the process discussion. It goes far beyond what can be constrained by the authors' new data and bits of it seem to me to be unfounded speculation.

278-280 Both of these authors were referring to glacial debris flows, at least in the case of Aksu and Hiscott confirmed by high resolution seismic and cores. This discussion is about muddy turbidity currents. Don't mix apples and oranges.

288-290 Again, apples and oranges. Debris flows have a fundamentally different flow mechanism from a fully turbulent muddy turbidity current.

291-305 It is not clear what point this paragraph is trying to make. It has long been known that turbidite systems sort sediment, both by segregating coarser sands from very fine sand-silt (e.g. between channels and levees) and by distal separation of fine-medium silt beds from muds with increasing higher proportions of clay size material (e.g. Piper, 1978, Fig. 12-7).

Figure 7. I understand the difficulty in drafting turbidity currents onto a block diagram. But this figure is quite misleading by showing a "meltwater turbidite" restricted to a channel. The whole point about the apparently aggrading levees in systems like this is that the turbidity currents appear not to "see" the channels and deposit uniformly over the levees. The data of Leng et al. (2018) suggest flow thicknesses in excess of 1 km.

385 Some are related to subglacial outbursts (e.g. Piper et al., 2007). But the new data of Gil and Keigwin (2018 EPSL) shows that mud accumulating intervals on Laurentian Fan are of longer duration and are thus unlikely to relate to oversized or exceptional outbursts.

This is certainly an important re-evaluation of the North Sea Fan. What is less clear to me is whether it represents anything more than an incremental improvement of our understanding of watery mid-latitude glacial outlets generally.

POINTS OF DETAIL

27 surely it is the data, not the model, that supports this claim?

34 references in chronological order

292 documented or interpreted?

David J.W. Piper
david.piper@canada.ca

Reviewer #2 (Remarks to the Author):

Dear Authors, I congratulate for having deeply revised the manuscript addressing all the comments provided with the previous reviews. In my opinion there is only one minor point I want to make:

Use of the term 'glacial channel'

Line 176

Line 212

Caption of figure 4, 5 (not I the labels)

Line 252

Line 262

Line 335

I have checked the scientific literature including the Atlas of Glacial Submarine Landforms

(Geological Society Special Publication 46, 2016) and the recent chapter 'Submarine Glacial Landforms' of the Book 'Submarine Geomorphology' (Springer, 2018). Also, the two papers cited in your manuscript when describing the channels: Rebesco et al., 2016 and Dowdeswell et al., 2018). In the absence of a standardised nomenclature of glacial marine landforms, the term 'glacial channel' is used for channels forming beneath the ice sheet, and in front of it (where they form glacial lakes) in continental or ice-proximal environments. In submarine geomorphology on continental slope environment I see a widespread the use of 'channel' without the modifier 'glacial' in front.

I note that in the illustrations the authors label the channels as 'channel' while the term 'glacial channel' is only used in the text.

My suggestion is therefore to modify the text avoiding the use of 'glacial channel' to be replaced by 'channel'. There is no impact on the sedimentary processes as presented in the paper.

BTW: Line 533 Rebesco et al., 2016. Page number is wrong. Should be 373-374.

Regards
Angelo Camerlenghi

Reviewer #3 (Remarks to the Author):

This is a re-review of the manuscript NCOMMS-19-30736A 'Meltwater Sediment Transport as the Dominating Process in Mid-latitude Trough Mouth Fan Formation'.

As mentioned in my previous review, this is an impressive dataset that is well presented. My main concern with the original version of the manuscript was the lack of discussion and critical review of the existing literature of TMF formation. Furthermore, the original version did not link the results/interpretation of this study back to the existing literature and the wider context was not explicit. The authors have been diligent with their corrections and with their response to reviewers and the authors have now addressed my comments in this revised version. The addition of discussion both in the introduction and the newly added 'Implications for Trough Mouth Fans' is welcome. I would advise that this manuscript be accepted for publication.

Reviewer 1 (David Piper)

- 1) Aksu and Hiscott (1992) and Mohrig et al. (1999) discussed transport in glacial debris flows, which have a fundamentally different flow mechanism from a fully turbulent muddy turbidity current.**

We agree that these references are dealing with debris flows, but this is also mentioned in the sentences. However, we now added three references and a sentence on muddy turbidite systems (lines 231-232) to show that this type of turbidite behavior (long runout, non-erosive, ...) is also documented in other regions of the Earth: Khripounoff et al. (2003) from the Zaire submarine valley, Carter et al. (2012) from Gaoping Canyon offshore Taiwan, and Azpiroz-Zabala et al. (2017) from Congo Canyon. However, this is the first study with extensive evidence (17,000 km² of high-quality 3D seismic data) showing that type of turbidite behavior on a glacial fan.

- 2) It is not clear what the point of paragraph in line 240-253 is trying to make.**

Our data suggest meltwater turbidites as the dominating process for the deposition of 450 m of glacial sediments during the last glaciation, and therefore grain-size fractionation of unsorted glacial sediments might also be present in trough mouth fans. This suggestion highlights that the sediments might laterally vary, from fine-grained clay in the levees to coarse-grained sands in the channels. This process has not yet been described on such a large scale as our evidenced in our study (extent of data and resolution of data). Grain-size fractionation has been modelled in numerical simulations and has been identified in sediment cores but has to our knowledge never been identified on such an extensive 3D seismic dataset. Previous studies usually make conclusions based on 2D seismic data or point measurements (sediment cores). The paragraph gives further information on what the uninterpreted sequences (seismic facies below resolution) might represent. For these reasons, we gave grain-size fractionation one paragraph of the manuscript.

- 3) Figure 7 is misleading by showing a “meltwater turbidite” restricted to the channel.**

We agree with the reviewer that meltwater turbidites are not all restricted to the channels. This point has been highlighted by a new meltwater turbidite flowing down to the right of the figure. However, we suggest that some turbidites followed existing channels. A block diagram is furthermore always a simplification. For the outcoming of this study, it is important to highlight that the reflections interpretable in the high-quality 3D seismic data have channel-levee expressions in planar view. The type of process how these channel-levee systems have been formed is an interpretation of the imaged structure maps. Therefore, the sketch shows quite some existing channels.

Reviewer 2 (Angelo Camerlenghi)

- 1) I have checked the scientific literature including the Atlas of Glacial Submarine Landforms (Geological Society Special Publication 46, 2016) and the recent chapter 'Submarine Glacial Landforms' of the Book 'Submarine Geomorphology' (Springer, 2018). Also, the two papers cited in your manuscript when describing the channels: Rebesco et al., 2016 and Dowdeswell et al., 2018). In the absence of a standardised nomenclature of glacial marine landforms, the term 'glacial channel' is used for channels forming beneath the ice sheet, and in front of it (where they form glacial lakes) in continental or ice-proximal environments. In submarine geomorphology on continental slope environment I see a widespread the use of 'channel' without the modifier 'glacial' in front. I note that in the illustrations the authors label the channels as 'channel' while the term 'glacial channel' is only used in the text. My suggestion is therefore to modify the text avoiding the use of 'glacial channel' to be replaced by 'channel'. There is no impact on the sedimentary processes as presented in the paper.

We agree with the reviewer and removed the term "glacial" from the channel morphologies discussed.

Reviewer 3

Reviewer 3 is satisfied with the improvements and did not recommend any corrections.